# Therapeutic Potentials of Immunometabolomic Modulations Induced by Tuberculosis Vaccination

**DOI:** 10.3390/vaccines10122127

**Published:** 2022-12-12

**Authors:** Bhupendra Singh Rawat, Deepak Kumar, Vijay Soni, Eric H. Rosenn

**Affiliations:** 1Center for Immunity and Inflammation, Rutgers New Jersey Medical School, Newark, NJ 07103, USA; 2Department of Zoology, University of Rajasthan, Jaipur 302004, Rajasthan, India; 3Division of Infectious Diseases, Weill Department of Medicine, Weill Cornell Medicine, New York, NY 10065, USA; 4School of Medicine, Tel Aviv University, Tel Aviv 6997801, Israel

**Keywords:** metabolomics, Bacillus Calmette-Guerin, *Mycobacterium tuberculosis*, tuberculosis, vaccinomics, vaccine metabolism, immunometabolism, trained immunity, epigenetics and immunity

## Abstract

Metabolomics is emerging as a promising tool to understand the effect of immunometabolism for the development of novel host-directed alternative therapies. Immunometabolism can modulate both innate and adaptive immunity in response to pathogens and vaccinations. For instance, infections can affect lipid and amino acid metabolism while vaccines can trigger bile acid and carbohydrate pathways. Metabolomics as a vaccinomics tool, can provide a broader picture of vaccine-induced biochemical changes and pave a path to potentiate the vaccine efficacy. Its integration with other systems biology tools or treatment modes can enhance the cure, response rate, and control over the emergence of drug-resistant strains. *Mycobacterium tuberculosis* (*Mtb*) infection can remodel the host metabolism for its survival, while there are many biochemical pathways that the host adjusts to combat the infection. Similarly, the anti-TB vaccine, Bacillus Calmette-Guerin (BCG), was also found to affect the host metabolic pathways thus modulating immune responses. In this review, we highlight the metabolomic schema of the anti-TB vaccine and its therapeutic applications. Rewiring of immune metabolism upon BCG vaccination induces different signaling pathways which lead to epigenetic modifications underlying trained immunity. Metabolic pathways such as glycolysis, central carbon metabolism, and cholesterol synthesis play an important role in these aspects of immunity. Trained immunity and its applications are increasing day by day and it can be used to develop the next generation of vaccines to treat various other infections and orphan diseases. Our goal is to provide fresh insight into this direction and connect various dots to develop a conceptual framework.

## 1. Introduction

Before the COVID-19 pandemic, Tuberculosis (TB) an infection caused by *Mycobacterium tuberculosis* (*Mtb*) was the leading cause of death from a single infectious agent ranking above both Human Immunodeficiency Virus (HIV) and malaria. The situation has become even worse amid the COVID-19 pandemic which has disrupted the global health infrastructure; causing the number of deaths due to TB to rise for the first time in over 20 years [1,2]. Bacterial drug resistance is a growing concern as even newly introduced anti-TB drugs such as bedaquiline and delaminid are failing to fight the infection [3]. Long treatment regimens, rising cases of HIV-TB coinfection, and the rapid emergence of drug-resistant strains underscore the necessity for newer prophylactic and therapeutic modalities with less or no dependency on antibiotics. Inhibiting the transmission cycle of TB with novel effective vaccines have the potential to achieve the World Health Organization’s (WHO) ambitious goal of ending the TB epidemic by 2035 [4].

To date Bacillus Calmette-Guerin (BCG) is the only licensed vaccine against TB. First used in 1921, it was orally administered to a child in Paris by Dr. Benjamin Weill-Halle [5]. Although BCG vaccination in infants is moderately effective in preventing severe, extrapulmonary TB, it has shown varied efficacy in preventing TB in adolescents and adults in different clinical trials [6]. It is unclear what different physiological and metabolic factors account for these variable immune responses to the BCG vaccine.

Several studies have revealed extensive alterations in the host metabolism upon *Mtb* infection [7,8] and subsequent treatment [9,10] (Table 1). These changes represent not only the state or progression of the diseases but also the success of the therapeutic interventions. Analysis of host-directed metabolomics can profile the non-genomic and non-transcriptional aspects of these by providing a snapshot of the metabolites released in body fluids such as blood or urine. It has been used to screen specific metabolic identifiers of hypoxic metabolism and inflammation during *Mtb* infection [11,12,13,14,15].

A distinct metabolic reprogramming is observed in response to demands caused by various bacterial or viral infections as well as vaccinations [16]. Glycolysis, fatty acid biosynthesis, bile acid metabolism, sphingolipid, and sphingomyelin production, different amino acid metabolism and catabolism, creatine metabolism, and glutaminolysis are common metabolic pathways essential for the immune system’s response against *Mtb*, *E. coli*, influenza, *SARS-CoV-2*, chikungunya, and dengue (Figure 1) [15].

It is well known that our immune system develops cellular memory upon infection and vaccination. However, the process by which this memory helps to protect against future exposures to unrelated pathogens is not well understood. A recent study observed the activation of innate immunity in an experiment where β-glucan (a component of *C. albicans* cell wall) challenged human monocytes and showed an ex vivo increase in cytokine synthesis in response to an unrelated stimulation [17,18]. There are some studies describing similar relationships of BCG vaccination with non-specific protection against different causes of mortality. This process of innate immune system adaptation was found to be associated with epigenetic changes in immune cells and is known as trained immunity [19,20]. A few studies have illuminated the effect of the BCG vaccine on host metabolism and its relationship with trained immunity. Specifically, changes in central carbon metabolism and lipid metabolic pathways have been linked to histone modification enzymes. In this review, we summarize the metabolic reprogramming that occurs due to the BCG vaccination, as well as its relationship with trained immunity, and strategies for the development of novel vaccines and applications. This discussion aims to describe TB pathogenesis and provide a condensed set of understandings of the host immunometabolism upon *Mtb* infection and vaccine administration (Figure 1).

**Table 1 vaccines-10-02127-t001:** Metabolic changes upon *Mtb* infection and BCG vaccine administration.

Metabolic Modulator	Sample Type	Metabolomics Technique Used	Affected Metabolites or Pathways	References
*Mtb* Infection	Blood serum, plasma, and urine	Gas chromatography-mass spectrometry, Liquid chromatography-mass spectrometry, Flow injection analysis-tandem mass spectrometry	Urea, sphingolipid, sphingosine-1-phosphate, sulfoxymethionine, fatty acid metabolism, sphingomyelins, phosphatidylcholines, lysophosphatidylcholines, amino-acyl tRNA, lysosome pathways, mannose metabolism, pyruvate, citrate, protein digestion pathways, asparagine, aspartate, citrulline, cysteine, lysine, leucine, methylamine, gamma-glutamylglutamine, glutamate, formate, glutamine, histidine, inosine, methionine, tryptophan, kynurenine, lactate, fatty acid beta-oxidation, itaconate, mycolic acids, phthiocerol dimycocerosate, glycerophosphocholine, nicotinate.	[7,8,21,22,23,24,25,26,27]
BCG Vaccination	Blood serum	Liquid chromatography-mass spectrometry	Purine biosynthesis, N6-carbomoyltheronyladenosine, glucose metabolism, alpha-ketobutyrate, 1,5-anhydroglucitol, methylguanine, fumarate, glutamate, glutamine, acetyl-CoA, lactate, glucose, nicotinamide adenine dinucleotide, 2-sulfotrehalose, trehalose 6-phosphate, mycobactin, 1-tuberculosynadenosine, conjugate-mycothiolhexadecanoyl-sn-glycero-3-phospho-(1′-myo-inositol), Hypoxanthine, para-aminobenzoic acid, lysophosphatidylcholines, lysophosphatidylethanolamines, sphingolipid metabolism, docosahexaenoic acid.	[9,10,14,28,29]

## 2. *Mtb*: A Smart Pathogen

*Mtb* has coexisted with human hosts since our early hominid ancestors over 3 million years ago. The long period of co-evolution between *Mtb* and humans has led to the development of complex molecular interactions between the pathogen and the host’s immune system [30]. *Mtb* is admitted into alveolar macrophages via host C-type lectin, CD91, and membrane Toll-like receptors [TLRs], which recognize the mycobacterial molecular patterns, such as cell wall lipoglycans and lipoarabinomannan [31]. *Mtb* not only evades the body’s primary defenses against pathogen entry but also survives within-host immune cells. Inside macrophages, *Mtb* prevents normal processes, such as phagolysosome formation and antigen presentation [32], by secreting virulence proteins that affect host pathways like the endosomal sorting complex. *Mtb* virulence factors (such as ESAT-6, phenolic glycolipid, and phthiocerol dimycocerosates) further drive macrophage polarization [33]. Upon stimulation by pathogen exposure macrophages shift metabolic pathways, allocating recourses to attack the pathogen. Understanding the molecular and metabolic reprogramming occurring in *Mtb* and the host cells can enhance our understanding of *Mtb* survival techniques; and improve therapeutic development. *Mtb* employs multiple strategies to survive inside the macrophage. It first avoids phagosome digestion by interfering with phagosomal maturation markers, such as Rab GTPase, involved in membrane fusion events [34,35,36]. This can also prevent the acidification required for the activity of lysosomal enzymes [37,38,39]. Studies report that *Mtb* not only survives inside the macrophage but ensures contact with the cytosol using the ESX-1/T7S and phthiocerol dimycocerosates (DIM/PDIM) systems [40,41,42,43,44]. This cytosolic accessibility is an important aspect of the complex host-pathogen interaction as it provides the access to essential nutrients [45,46]. A better understanding of this cytosolic exposure mechanism of the pathogen can help to improve the protective efficacy of vaccine candidates. *Mtb’s* interference with lysosomal enzymes and host metabolism may also impact host cell apoptotic and necrotic pathways, further contributing to *Mtb* survival, replication, and dissemination [44,47,48,49].

*Mtb* infection can result in granuloma formation, wherein the bacterium is sequestered and prevented from causing infection, but it is also protected from host immune destruction [50]. *Mtb* displays self-regulated epigenetic changes when entering the latent stage, allowing for adjustment of metabolism and survival in low pH [51]; permitting *Mtb* to persist within the granuloma. Each site of granuloma formation provides a microenvironment that induces phenotypic and metabolic heterogeneity. Cavitation, when a granuloma penetrates the underlying tissue, can allow the pathogen to access oxygen and a privileged environment [52]. Relative adaptation to these stringent environments results in increased pathogenicity during secondary reactivation [53]. It has been reasoned that this plasticity is a result of *Mtb*’s long period of co-evolution with humans. Through the duration of the latent infection, CD4^+^ T-cells specifically produce a Th1 cytokine reaction that functions to maintain granuloma integrity. Later during TB infection, a more significant role is played by CD8^+^ T-cells, which regulate the lysis of infected cells. T-cells also activate macrophages by releasing Tumor Necrosis Factor Alpha (TNF-α) and Interferon gamma (IFN-γ) [54].

## 3. Host Metabolic Adaptation upon *Mtb* Infection

Along its timeline of evolution, *Mtb* has learned to survive within the human host through fine-tuning metabolism and counteracting the host immune system. Gluconeogenesis is the primary source of carbon for the bacterium in the host. Phosphoenolpyruvate carboxykinase (PEPCK) was found to play an essential role in bacterial survival during macrophage infection [55]. With the help of isotope labeling and metabolomic profiling, it was found that *Mtb* can simultaneously utilize multiple types of carbon substrates (such as glycerol, acetate, and dextrose) for sustained growth [21] (Table 1). *Mtb* favors fatty acid metabolism during the infection but can rely on many metabolic pathways [56,57]. The serum metabolome shows higher amounts of aspartate, sulfoxy methionine, and glutamate and lower levels of asparagine, methionine, and glutamine in patients with active TB as compared to patients with latent infections or healthy subjects [8]. Macrophages overexpress HIF-1α (hypoxia-inducible factor-1) which upregulates lactate dehydrogenase-A (LDH) and lowers pyruvate levels, thus reducing the availability of glucose for *Mtb* survival [7] (Table 1). Targeted metabolomics of human plasma has revealed that both active and latent *Mtb* enhance the catabolism of tryptophan to kynurenine, and it is reversed with the anti-TB treatment [22]. Increased tryptophan metabolism was linked to a CD4^+^ T cell response promoting immune tolerance and protecting the host from inflammation bursts [58]. Host phospholipase D was also found to play a key role in controlling the *Mtb* infection via Sphingosine 1-phosphate (S1P) metabolism [59] (Table 1).

## 4. BCG Vaccine: Immunometabolic Reprogramming and Trained Immunity

Calmette and Guerin first used a live-attenuated *Mycobacterium bovis* strain along with beef bile and found this resulted in protection against TB in bovine and rodent animal models [60,61]. Genomic studies further revealed that knocking out the region of deletion 1 (RD1) was associated with loss of *Mtb* virulence. This attenuated strain was termed BCG. RD1 is a 10.7 kb fragment containing 9 Open Reading Frames (ORFs) normally present in virulent *Mtb* and *M. bovis* strains. Genes present in the RD1 region encode for two major proteins of *Mtb*, ESAT-6, and CPF-10, parts of ESX1 secretion system. Although the non-virulent phenotype is predominantly associated with the deletion of the RD1 region, however, its reintroduction does not completely restore the virulence suggesting the role of other factors [62].

Heterogeneity between individuals causes some variation in the trained immune system and responses to BCG vaccination, however, the basis for these differences has not been well defined [63,64]. The composition of circulating metabolites impacts both innate and adaptive immune components and might contribute significantly towards variations observed in the immune responses to vaccines among individuals.

Many metabolites are associated with trained immunity. Particularly BCG vaccination led to an increase in glycolysis and glutamine metabolism (Figure 2, Table 1). Conversely, inhibiting glutaminolysis reduces the trained immune responses [65]. Fumarate is another key metabolite that accumulates in the primed cells following glutaminolysis. Specifically, it has been shown to facilitate trained immunity by inhibiting KDM5 histone demethylase activity and enhancing H3K4me3 at the promoters of immune genes [66]. BCG-induced trained immunity has been implicated in modulating the plasma concentration of succinate and malate, key intermediates of the TCA cycle, and glutamine and glutamate metabolism [10] (Table 1). Furthermore, succinate is implicated in innate immune signaling and promoting increased production of IL-1β in macrophages [67]. Overall, these studies highlight the critical role of metabolites in regulating the trained immune responses following BCG vaccination (Figure 2).

The rapid protection by BCG in newborns suggests it might be mediated by the induction of innate immune responses. The different mechanisms involved in BCG-induced protection are being actively investigated [68,69,70]. In a recent study involving 6544 high-risk neonates, it was found that BCG induces granulocyte colony-stimulating factor (G-CSF) to promote granulopoiesis resulting in the enhanced generation of neutrophils that protect from neonatal sepsis [71].

One potential mechanism for BCG-induced heterologous protection is the epigenetic reprogramming of innate immune cells, a process termed trained immunity (Figure 2) [19,20,72]. Through this process, innate immune cells such as macrophages, monocytes, and natural killer (NK) cells, are prepared for rechallenge with heterologous stimuli later in life (Figure 2) [19,20,72]. The changes induced by BCG exposure affect immune cell responsiveness, resulting in their enhanced effector functions, such as the release of cytokines and reactive oxygen species upon encountering non-related infections. BCG-associated protective effects might be driven by the modulation of metabolites [73]. These metabolites may serve as cofactors for enzymes involved in the modification of chromatin and DNA to train the immune cells [72]. The active metabolites needed for maintaining trained immune responses are derived from the modulation of glucose, glutamine, and cholesterol metabolic pathways [74]. Diverse factors including physiological or pathological states of individuals and environmental factors such as nutrition, impact the human metabolome [75,76].

The differences in active or dysregulated metabolic pathways identified in healthy and diseased individuals are being extensively categorized. Compared to other basic research, the mechanistic insights yielded through a metabolomic approach focus more on the gene- interactome intersection. By nature, these are key regulatory points with high therapeutic potential [76,77]. There is a reciprocal relationship between plasma metabolite levels and the immune response. Plasma composition can modulate immune activation, while systematic immune responses can affect metabolism and thus plasma metabolite levels. Hematologic and other modes of metabolite quantification are emerging as novel tools in disease staging and possibly prediction of pathologic progression [15,78].

As discussed above the BCG vaccine is generally given earlier in life as it shows less protective effects when administered in adolescence and adulthood. In infants, several metabolites in monocytes and macrophages, including fatty acids, acetyl-coenzyme A, and succinate, regulate the epigenetic modulations required for trained immunity [64,79,80,81]. In recent years, mass spectroscopy-based metabolomics has been utilized to examine neonatal plasma [82] however, few studies have used this technique to examine the immune response to vaccines in infants [83,84,85]. In another clinical study (*n* = 100), metabolomic profiling of infant plasma demonstrated that BCG vaccination induces a metabolic shift in lysolipid pathways, including lysophosphatidylcholines [9] (Table 1). Significantly increased levels of some monoacylglycerols, sphingolipids, steroids, and lipoprotein lipase metabolites along with decreased biosynthesis of progestin steroids and palmitoylglycerols, were also observed upon BCG vaccination [9]. While recent research has shed light on the complexity of metabolic shifts during the immune response, more investigation toward understanding neonatal immunometabolism is required [86].

## 5. Can BCG Be Used for Other Diseases?

In addition to protecting against tuberculosis, BCG vaccination is associated with enhanced protection from some unrelated infections and leads to reduced mortality. Interestingly, upon BCG administration, healthy human subjects showed a phenomenon similar to the cytokine storm, which persisted for almost 3 months after the vaccination [19,87]. BCG can also trigger heterologous immunity (antigen-independent) characterized by IL-17 and INF-γ productions from T-cells [88]. Studies have demonstrated that the presence of a BCG scar or PPD (purified protein derivative) is associated with a significant reduction in infant mortality [89,90]. Further research using randomized trials concluded the reduction of neonatal mortality associated with early BCG-vaccination may be a result of decreased incidence of respiratory infections and sepsis [91,92,93]. BCG vaccination was also found to have protective effects against lethal candidiasis in regular mice [94], and severe combined immunodeficient (SCID) mice, suggesting a role of B and T cell-independent immune response [19].

A study by Arts, R.J.W. et al. revealed a significant association of BCG with non-specific production of IL-1β and protection against yellow fever viremia [64]. Subsequently, it is known to offer protection against the Influenza virus [95], *Candida albicans* [19], Leishmania species [96,97], malaria [63], human immunodeficiency virus (HIV) [98], and hepatitis C virus (HCV) [99] infections. Additionally, this may occur due to BCG-mediated modulation of T-cell-directed autoimmunity [100]. Moreover, BCG was found to be effective in bladder cancer. Various clinical trials show that intravesical injection of BCG can control the recurrence of superficial bladder tumors by provoking local immune activity [101]. Since BCG affects the host glucose metabolism, its effect on Type 1 diabetes mellitus (T1DM) was also studied by many groups. Shehadeh et al. administered 17 T1DM patients with a single dose of BCG and found that 65% of subjects show clinical remission by 4th week [102]. While many other clinical trials failed to establish the correlations between BCG and T1DM [13], they all were done before the year 2000, when metabolomics was not well developed. Because BCG affects the host-trained immunity and metabolism, it has also been proposed as a treatment for asthma, allergic rhinitis, and atopic dermatitis. When administered via the intranasal route, BCG significantly attenuated allergy-related inflammation and hypersensitivity [103,104]. However, enough significant epidemiological and metabolomic data are not yet available to establish a relationship between BCG and atopy.

### Novel TB Vaccines

The target of WHO’s ‘End TB’ program to achieve a 95% reduction in deaths from TB and a 90% reduction in TB incidence by 2035, requires multisector efforts incorporating socio-economic identity, novel diagnostic and therapeutic interventions, and effective vaccine development. Vaccines have shown their potential in controlling and even eradiating many life-threatening diseases. For this reason, the End TB program has an immediate need for a vaccine candidate that is effective in all age groups, for preventing infection, improving antibiotic treatment outcomes, and reducing relapse events. Current TB vaccine candidates are grouped into several categories including viral vector vaccines, attenuated live vaccines, inactivated whole cell vaccines, and adjuvanted protein subunit vaccines, among other formulations.

Examples of recombinant viral vector vaccines include the Ad5 Ag85A, MVA85A, ChAdOx1 85A, and TB/FLU-04L engineered strains (Table 2). The Ad5 Ag85A vaccine in particular is designed as a replication-deficient adenoviral serotype 5 (Ad5) vector containing the Ag85A antigen from *Mtb* (Table 2). Typically, this strain is incorporated in the booster vaccine given after BCG priming [105,106,107,108]. MVA85A (recombinant modified vaccinia virus Ankara containing antigen 85A from *Mtb*) was reported to induce a robust Ag85A-specific CD4^+^ and CD8^+^ response. In animal cell models incubated with low-dose aerosol infections, this immunologic modem displayed increased protection compared to those induced with BCG [109]. These same studies have described the chimpanzee adenovirus [110] as expressing the Ag85A antigen of *Mtb* (ChAdOx1 85A) [111]. The corresponding mucosal vector vaccine composed of a replication-deficient attenuated influenza virus expresses Ag85A and ESAT-6 *Mtb* antigens (TB/FLU-04L), the protective efficacy of BCG (Table 2) [112].

VPM1002 and MTBVAC are viable whole-cell vaccines that can carry multiple antigens (Table 2). Like BCG, these strains also have complex and individualized immune reactions. As observed in the case of BCG, pre-sensitization by these nontuberculous mycobacterial strains reduces the pathologic consequences of subsequent mycobacterial infections. VPM1002 is developed as a recombinant BCG vaccine with improved efficacy. The immunogenicity of this vaccine was tempered with genetic engineering techniques, modulating the release of mycobacterial antigens into the host cell cytosol. Specifically, the gene encoding the urease C (UreC enzyme) was replaced with the Listeriolysin O (LLO) encoding sequences (Hly) of *Listeria monocytogenes* (Table 2) [113,114]. Normally the urease C enzyme inhibits phagocytic lysosome maturation and improves the survival of *Mtb* in macrophages [115,116]. The other possible vaccine candidate, MTBVAC is a live, attenuated *Mtb* strain derived from an *Mtb* clinical isolate belonging to modern lineage 4 which retains the T cell epitopes described in tuberculous mycobacterial strains, as well as the ESAT6 and CFP10 antigens. In preclinical evaluation, MTBVAC showed improved efficacy as compared to BCG (Table 2) [117]. Attenuation was achieved by the deletion of two genes, *phoP,* and *fadD26*, a crucial modulator of virulence [117]. Currently, the MTBVAC formulation is being adapted for use as a preventive vaccine as well as a booster vaccine for BCG-primed adults.

RUTI, Vaccae, DAR-901, and Immuvac are vaccine candidates based on inactivated whole-cell derivatives (Table 2). RUTI is a multi-antigen vaccine derived from the cell wall of *Mtb* cultivated in hypoxic conditions and is formulated as a liposomal suspension (Table 2) [118]. Vaccae is derived from heat-killed *Mycobacterium vaccae*, a non-pathogenic environmental mycobacterium. It was reported to complement the effectiveness of BCG vaccination, resulting in an enhanced immune response against TB (Table 2) [119]. Similarly, DAR 901 is derived from heat-killed non-tuberculous mycobacteria identified from the SRL-172 master cell bank [120]. Clinical trials examining the efficacy and safety of the DAR-901 vaccine in BCG-immunized adults determined that it effectively induced the formation of cellular and humoral immune responses while displaying no adverse effects after three doses (Table 2) [121,122,123]. Immuvac, is a vaccine derived from heat-killed *Mycobacterium indicus pranii* a non-pathogenic mycobacterium closely related to *M. avium* [124]. It is currently approved by the Central Drugs Standard Control Organization, India (CDSCO) and the FDA as an immunotherapeutic and immunoprophylactic agent for treating multibacillary leprosy [125,126,127].

Multiple protein subunit vaccines have also been developed including AEC/BC02 (using Ag85B and ESAT6-CFP10 antigens with BC02 compound adjuvant) [128]; H56:IC31 (using Ag85B, ESAT-6, and *Rv2660c* antigens with IC31 adjuvant) [129]; ID93 + GLA-SE (using Rv2608, Rv3619, Rv3620 and Rv1813 antigens with GLA-SE adjuvant) [130], and M72/AS01E (using Mtb32 and Mtb39 antigens with AS01B, AS02A, or AS01E adjuvant) [131,132,133,134,135] (Table 2). Protein subunit vaccines are primarily developed for use as booster vaccines in BCG-primed individuals to extend immune protection.

Despite all the efforts in this area the goal of ending the TB epidemic is still far away. During 2015–2021 the total incidence of TB infection and mortality was reduced by 10% and 5.9%, respectively. This is modest considering the 50% and 75% reduction milestones for infection and mortality set for 2025 by the WHO (Global tuberculosis report, WHO, 2022). The development and introduction of newer technologies are needed to fuel research in the TB diagnostic and therapeutic areas. While the whole genome sequence (WGS) of the *Mtb* (H37Rv) was uncovered in 1998 [136], recent developments in genomic techniques have provided a clear resolution at the nucleotide level. Whole genome analysis allows for screening clinically relevant genetic information, such as sequences related to drug resistance, and variations between bacterial species and strains [137]. Additionally, these techniques can help identify specific genetic regions responsible for antigenicity and virulence.

The TB vaccine development space can be improved by enhancing our understanding of the host immune response to TB infection and finding an effective route of antigen delivery (immunization) leading to sensitization [138]. Nanomaterial-based vaccine delivery systems have also shown potential in vaccine storage, improved stability of antigens in blood, and greater specificity in targeted delivery [139,140]. Rapidly emerging mRNA-based vaccination technology has shown its potential in vaccine development against many diseases including SARS-CoV-2 [141]. Current research in this area is focused on improving mRNA stability, optimizing delivery vectors, and enhancing control of protein expression. These advances have already led to the development of novel therapeutics such as self-amplifying RNA vaccines [142]. A recently adapted *Mtb* subunit vaccine, ID93, is composed of a self-replicating RNA molecule with a nanostructural carrier [143].

**Table 2 vaccines-10-02127-t002:** TB vaccine candidates with clinical trial status.

Vaccine Category	Vaccine Candidate	Antigen and Formulation	Latest Clinical Trial Phase (Status) ^#^	NCT Number (References)
Recombinant viral vector	Ad5 Ag85A	Ag85A antigen expressed in Adenovirus serotype 5	I (Completed in 2021)	NCT02337270 [144,145,146]
	MVA85A	Ag85A antigen expressed in modified Vaccinia virus Ankara	IIa (Completed in 2021)	NCT03681860 [109]
	ChAdOx1 85A	Ag85A antigen expressed in Chimpanzee adenovirus	I (Completed in 2021)	NCT03681860 [110,147]
	TB/FLU-04L	Ag85A & ESAT-6 antigens expressed in attenuated replication-deficient influenza virus vector	I (Completed in 2015)	NCT02501421 [148]
Viable whole-cell	VPM1002	Recombinant BCG vaccine	III (Ongoing)	NCT04351685 [149,150]
	MTBVAC	Attenuated *Mtb* clinical isolate with ESAT6 & CFP10 and independent genetic deletions of *phoP* & *fadD26* genes	II (Completed in 2022)	NCT03536117 [151,152]
Inactivated whole-cell	RUTI	Polyantigenic liposomal formulation of detoxified, fragmented *Mtb*	II (Ongoing)	NCT04919239 [153,154,155]
	Vaccae	Heat-killed *M. vaccae*	III (Completed in 2017)	NCT01979900 [156]
	DAR-901	Heat killed nontuberculous mycobacteria	II (Completed in 2020)	NCT02712424 [121,122,157,158]
	MIP/Immuvac	Whole cell, heat inactivated *Mycobacterium indicus pranii*	III (Completed in 2012)	NCT00341328 [159]
Protein subunit	AEC/BC02	Ag85b, ESAT6-CFP10 fusion protein, with BC02 adjuvant	II (Ongoing)	NCT05284812 [128]
	H56:IC31	Fusion protein of Ag85B, ESAT-6 and *Rv2660c* with IC31 adjuvant	II (Ongoing)	NCT03512249 [160,161,162,163]
	ID93 + GLA-SE	Fusion protein of Rv1813, Rv2608, Rv3619, and Rv3620 with GLA-SE adjuvant	IIa (Unknown)	NCT03806686 [164,165,166]
	M72/AS01E	Fusion protein of Mtb32A and Mtb39A with AS01E adjuvant	II (Ongoing)	NCT04556981 [167,168]

^#^ Status of the respective clinical trial as shown on clinicaltrials.gov (accessed on 7 December 2022).

## 6. Key Challenges and Recommendations

With the increasing prevalence of infections by drug resistant *Mtb* strains, it is important to find alternative therapies. The integration of vaccinology and metabolomics has formed a key development pipeline. Due to its impact on immune and metabolic responses, BCG has been considered a therapeutic and prophylactic tool against various autoimmune and inflammatory diseases. However, it comes with many challenges. One outstanding problem is that metabolomic studies have mainly been limited to murine models, but the molecular mechanism of the actions of these effects are yet to be investigated. There are many enhancers like rapamycin (a modulator of the mTOR complex), which have been used along with BCG to boost immunogenicity in mice [169]. However, these need to be validated in clinical trials with bigger cohorts. Coadministration of some other metabolic modulators (for example Dichloroacetate, Metformin, Silybin, and 2-deoxyglucose) can also be useful to amplify the immune impact of the BCG vaccination [65].

Tuberculosis infection has heterogeneous pathology containing both active and latent bacilli. This makes complete eradication of the *Mtb* infection difficult [170]. Metabolomic understanding of the immune system is still in its infancy. In addition to genetics many non-heritable variables such as food habits, geographical location, previous disease or vaccination status, and other environmental factors can confound the observations of different clinical studies. To account for this, all therapeutic studies might be designed using a multifactorial approach and analysis [171]. Furthermore, combining other omics datasets with immunometabolomics information can reveal complex genomic, proteomic, or transcriptomic clusters, implicating their role in immune activity and vaccination.

### Future Prospective

Tuberculosis is an old disease however, its cure remains a major challenge. Advanced systems biology approaches present solutions to understand the disease and account for the complex host-pathogen relationship. Novel metabolomic and chromatographic technologies such as hydrophilic interaction chromatography (HILIC), ion-exchange chromatography, chiral liquid chromatography, capillary electrophoresis mass spectrometry (CE-MS), ion-mobility separation (IMS), and liquid-liquid micro extraction (LLME) can be helpful to analyze a broad range of metabolites with great accuracy [172]. Machine learning (ML) and artificial intelligence (AI) based programs are facilitating next-generation solutions for high-resolution mass spectrometry (HRMS) with improved data quality, accurate metabolite annotations, and network analysis [173]. With the emergence of the COVID-19 pandemic, the landscape of infectious disease research has changed. Vaccines like BCG, which have an impact on trained immunity and metabolism, provide an adjunctive approach to pandemic preparedness. We strongly believe that the formation of global shared databases can aid in the development of precision models of personalized vaccination against several life-threatening diseases. This collaborative effort could also help to allocate vaccine resources and to predict possible epidemiologic events well in advance.

## 7. Conclusions

The human metabolome is vast and is under the influence of many pathological and physiological factors. With the development of high-quality mass spectrometry and chromatography, metabolomics is evolving as a tool to study diseases and therapies. In the last few years, various studies have shown that metabolism is associated with the functional state of the human immune system. Metabolic factors can affect the protective natural immune response against diseases as well as the efficacy of vaccines. Knowledge of immunometabolism in diseased and vaccinated individuals presents an immense unharnessed therapeutic potential to cure many autoinflammatory diseases, viral and bacterial infections, and metabolic disorders.

BCG is one of the oldest vaccines in human history, saving millions of lives every year. By highlighting metabolic aspects of the BCG vaccine pertaining to the human immune system, our review presents metabolomics as a promising vaccinomics tool for discovering novel avenues to cure difficult infections. BCG impacts glycolysis in immune cells and activates lactate biosynthesis through mTOR and HIF-1α signaling. It was also found to affect the levels of TCA cycle intermediates, hence affecting lipid biosynthesis and the activity of histone-modifying enzymes (such as HAT and KDMs) via fumarate and succinate. These lead to epigenetic modifications and induction of trained immunity in different innate immune cells such as monocytes, macrophages, and NK cells. Upon subsequent exposure, these cells increase the relative secretions of proinflammatory cytokines and provide protection against many non-specific diseases and infections for which no therapy yet exists. Many unanswered questions remain, such as what frequency of BCG vaccination is required to sustain the trained immunity. Greater analysis is needed to uncover the exact role of metabolic modulators in trained immunity and the impacts of many unknown metabolites (the dark matter of metabolism). As we enter the era of Big Data, novel analysis methods such as machine learning are required to implement a multi-omics approach to understanding immunity. This understanding can aid the development of novel anti-TB vaccines, leading to better immunogenic vectors with wider applications. Combining such vaccines with other modes of treatment can allow for targeted therapy of many diseases and reduce the emergence of drug resistance. Cross-discipline collaborations and global data integrations can accelerate the development of metabolomics applications in vaccinomics, to make informed decisions in clinical settings.

## Figures and Tables

**Figure 1 vaccines-10-02127-f001:**
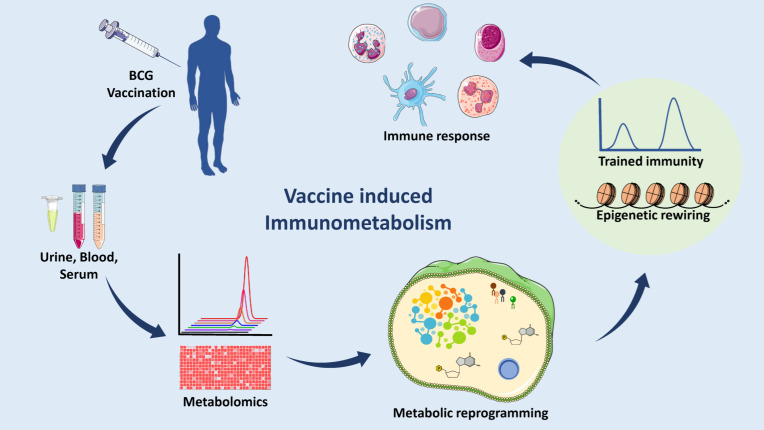
Analyzing Immunometabolomic effects of TB vaccination. In a clinical setup, body fluids (such as blood serum or plasma and urine) would be collected and processed to run on a GC-MS or LC-MS to profile the levels of the metabolites and then analyzed various bioinformatic tools. Quantitative analysis shows the changes in different pathways as compared to placebo subjects. A correlation has been shown between metabolic remodeling, trained immunity, and epigenetics. These may potentiate the effect of the vaccine to mount a better immune response. (Abbreviations: GC-MS: Gas chromatography-mass spectrometry, LC-MS: Liquid chromatography-mass spectrometry.) (The figure was partly generated using Servier Medical Art, provided by Servier, licensed under a Creative Commons Attribution 3.0 unported license).

**Figure 2 vaccines-10-02127-f002:**
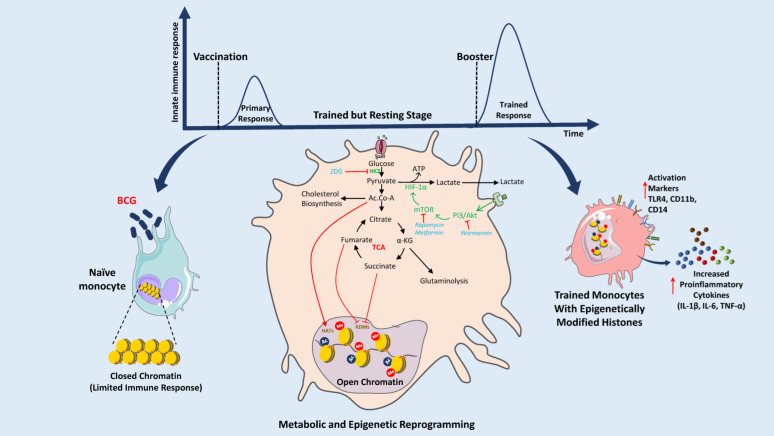
Mechanism of BCG-induced metabolic reprogramming and its role in the activation of trained immunity through epigenetic modifications of histones. BCG stimulates innate immune cells such as naïve monocytes, macrophages, and natural killer (NK) cells. Phagolysosomal digestion of BCG induces PI3/Akt and mTOR signaling pathways which directly affect the expression of HIF1α thus modifying the glycolysis and the TCA cycle. Acetyl-CoA (Ac.CoA), fumarate, and succinate can affect the activity of HATs and KDMs and induce epigenetic modifications of histones. These modifications of histones result in better access to promoters of proinflammatory cytokines (e.g., IL-1β, IL-6, TNF-α) and program immune cells for the next encounter. Booster or second exposure results in an excessive amount of cytokines and activation of different markers (TLR4, CD11b, and CD14). This trained immunity can protect against many diseases like sepsis, type 1 diabetes mellitus, pneumonia, asthma, allergic rhinitis, melanoma, etc. (Abbreviations: BCG: Bacillus Calmette-Guérin, PI3: Phosphatidylinositol 3-kinase, Akt: Serine/Threonine protein kinase B, mTOR: mammalian Target of Rapamycin, HIF1α: Hypoxia Inducible Factor 1 Subunit Alpha, ATP: Adenosine triphosphate, TCA: Tricarboxylic Acid cycle, IL-1β: Interleukin-1 beta, IL-6: Interleukin 6, TNF-α: Tumor necrosis factor alpha, TLR4: Toll-Like Receptor 4, CD11b: Cluster of Differentiation molecule 11B, CD14: Cluster of Differentiation 14, 2DG: 2-Deoxy-d-glucose, Ac.CoA: Acetyl coenzyme A, Me: Methylation, HAT: Histone Acetyltransferases, KDMs: Lysine Demethylases.) (The figure was partly generated using Servier Medical Art, provided by Servier, licensed under a Creative Commons Attribution 3.0 unported license.).

## Data Availability

There are no supporting data associated with this article.

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
