# Peer review of "Therapeutic Potentials of Immunometabolomic Modulations Induced by Tuberculosis Vaccination"

_vaccines, 2022, doi:10.3390/vaccines10122127_

Round 1
Reviewer 1 Report
Title:
Title is too short.
The aim of title is to capture the reader’s attention.
The title should adequately describe, the purpose, content and main findings of the research
Objectives:
Clearly describe General and Specific objectives of the study (too many study questions)
BCG vaccine:
Highlight variable effectiveness of the BCG vaccine against adult pulmonary TB, and vaccine potential interference with tuberculin skin test sensitivity
Mtb : a smart pathogen :
Describe ability of Mtb to evade our immune system and how Mtb manages to survive and multiply inside macrophages
Novel TB Vaccines:
Highlight the need of novel TB vaccine to end TB by 2030 to achieve global target
Describe how new approaches in vaccine development use novel biotechnology, target new mechanisms and shape the immune system response
Conclusion:
Needs improvement. Restate research problem, summarize your arguments and describe briefly the implications
Author Response
Title:
Title is too short.
The aim of title is to capture the reader’s attention.
The title should adequately describe, the purpose, content and main findings of the research
Response: Thanks for pointing out this. We have elaborated the title and changed it to “Therapeutic Potentials of Immunometabolomic Modulations Induced by Tuberculosis Vaccines”.
Objectives:
Clearly describe General and Specific objectives of the study (too many study questions)
Response: Thanks for the correction. We have removed multiple questions and added one specific point in the last paragraph of the introduction. This would help to visualize the simple objective of the review and convey the focused message.
BCG vaccine:
Highlight variable effectiveness of the BCG vaccine against adult pulmonary TB, and vaccine potential interference with tuberculin skin test sensitivity
Response: Thanks for pointing out this. We have added a few studies on this topic, however from metabolomic point of view.
Mtb : a smart pathogen :
Describe ability of Mtb to evade our immune system and how Mtb manages to survive and multiply inside macrophages
Response: Thanks again for the suggestion. Please check the first paragraph of the section headed “Mtb: A Smart Pathogen”.
Novel TB Vaccines:
Highlight the need of novel TB vaccine to end TB by 2030 to achieve global target
Response: Thanks for the suggestion. We have included a detailed paragraph (first, and sixth paragraph of “Novel TB Vaccine section). We hope that would suffice the requirements.
Describe how new approaches in vaccine development use novel biotechnology, target new mechanisms and shape the immune system response
Response: Once again, thanks for the suggestion. We have included a total detailed paragraph in “Novel TB vaccine” section. A table has also been included to summarize the suggested points.
Conclusion:
Needs improvement. Restate research problem, summarize your arguments and describe briefly the implications
Response: We apologize for not elaborating the conclusion. We have rewritten it completely with the key takeaways of the review. Please check the conclusion section.
Reviewer 2 Report
The paper is positioned as an attempt to review the advantages of immunometabolomic approach to anti-TB vaccine improvement. The authors put forward such question as: "Would it be possible to develop a better anti-TB vaccine just by using metabolic enhancers or suppressors?" Taking into account that BCG remains the only licensed anti-TB vaccine for 100 years, it is not surprising that the authors' question remains unanswered. The overrall impression of the review is a superficial approach to the analysis of complex phenomena. Such problems as Mtb evasion mechanisms and host adaptation to Mtb infection deserve much more detailed discussion. My recomendation to the authors would be to focus on exact metabolomic parameters which could be correlated with protective or thrapeutic properties of the vaccine.
Author Response
Recommendation: focus on exact metabolomic parameters which could be correlated with protective or thrapeutic properties of the vaccine
Response: Thanks for the suggestion. We have included a few more studies on this topic. Please check the 3rd and 4th and last paragraph in the “BCG vaccine: Immunometabolic Reprogramming and Trained Immunity” section. We have also included figure 2 for easy understanding.
Reviewer 3 Report
-Not well written. No searching strategy was used or mentioned. They reviewed these studies in these sections without providing details on: how many studies were included or met the selection criteria, how to extract and assess these studies, what are the key findings, etc.
Suggestion: if authors are willing to revise it by following the PRISMA guidelines and do a proper systematic review, this can be considered. In particular, they need to address the key areas such as number of studies included for analysis, key results or findings from this review, and study limitations, etc. Check the PRISMA guidelines for systematic review and meta-analysis.
- I would like to request youA table be included with manuscript. The table should summarize the contents of the paper in a concise.
- It is better for the authors to carefully edited by a fluent native English
- Several grammatical errors that make it very difficult to understand.
- It is better for the authors to revise and expand their conclusion
- New references for 2021, 2022 will be added to text
- Future aspects are missing
- The first time you include acronyms within the text, you have to write them in full. After that, you should report them as abbreviations only.
Author Response
No searching strategy was used or mentioned. They reviewed these studies in these sections without providing details on: how many studies were included or met the selection criteria, how to extract and assess these studies, what are the key findings, etc.
Response: Thanks for indicating this important point. We have included these details with respective studies.
Suggestion: if authors are willing to revise it by following the PRISMA guidelines and do a proper systematic review, this can be considered. In particular, they need to address the key areas such as number of studies included for analysis, key results or findings from this review, and study limitations, etc. Check the PRISMA guidelines for systematic review and meta-analysis.
Response: Thanks again for the suggestion. We have gone through the PRISMA guidelines and tried our best to incorporate them into our writing.
- I would like to request youA table be included with manuscript. The table should summarize the contents of the paper in a concise.
Response: Thanks for the suggestion. We have incorporated another figure (figure 2) and one table (table 2) to summarize the contents in a simple way.
- It is better for the authors to carefully edited by a fluent native English
Response: Thanks. Yes, we got it checked with a native English expert.
- It is better for the authors to revise and expand their conclusion
Response: Thanks for the suggestion. We have restructured the conclusion and explained it in detail. Kindly check the “conclusion” section.
- New references for 2021, 2022 will be added to text
Response: Thanks for the suggestion, we have incorporated many new references from the last 3-4 years. Kindly check.
- Future aspects are missing
Response: Thanks, we have incorporated “Future Prospective” section just before the conclusion, kindly check.
- The first time you include acronyms within the text, you have to write them in full. After that, you should report them as abbreviations only.
Response: We apologize for this. We have rechecked the manuscript and corrected the mistake.
Reviewer 4 Report
Known in the field based on previous literatures:
1. Tuberculosis (TB) is an infectious disease caused by the bacterial pathogen Mycobacterium tuberculosis and BCG is an avirulent tuberculosis strain historically given as vaccine to protect against tuberculosis.
2. Metabolomic studies using biofluids primarily urine, whole blood and serum, can be used along with omics to understand host–pathogen interactions at small-molecule levels. The interplay of immunology and metabolism- immunometabolism, has received increasing interest because of its role in the function and regulation of immune system processes in health and disease.
In this review authors discussed following findings:
I have gone through the review titled " An Immunometabolomic Perspective on Tuberculosis Vaccine’. This review gathered the information concerning metabolic relationships of BCG with the host immune system. Authors have mentioned following main points-
1. Authors discussed different anti-TB vaccine development in the review revealed some interesting questions and key challenges.
2. Authors also discussed the future application of metabolomics in anti-TB vaccine development.
The review presented are interesting and generally supportive of the conclusions drawn. The following minor suggestion if incorporated could help in the better understanding of the significance of the work and implications.
Minor Concerns:
1. As authors mentioned, there are several existing literatures were combining omics and metabolomic signatures significantly provide complementary information on TB progression. Explain, how your review is different from rest and how does it address a specific gap in the field?
Author Response
Minor Concerns:
- As authors mentioned, there are several existing literatures were combining omics and metabolomic signatures significantly provide complementary information on TB progression. Explain, how your review is different from rest and how does it address a specific gap in the field?
Response: Thanks for the kind words and the suggestion. We appreciate the valid concern. But when we are saying that there are many studies, we mean they are research work. Sorry for the confusion. There is no such updated review that explains the therapeutic implications of metabolic reprogramming mediated by BCG vaccination. We have reviewed research outcomes both from clinical and laboratory studies and explained that how it can be or can not be used for other diseases. Hope this satisfy the concern.
Reviewer 5 Report
The paper named “An Immunometabolomic Perspective on Tuberculosis Vaccine” is a well done review where the data are clearly exposed and easy to read. In this review author reviewed various metabolic consequences of Mtb infections and the potential of various host biochemical changes upon BCG vaccination. Moreover described the immunology of TB and how different immunometabolic signatures can be useful to potentiate the existing vaccine and new vaccine developments. At the end of the review author discussed the novel anti-TB vaccine development and try to give a vision about the future application of metabolomics in anti-TB vaccine development.
Only minor point is required in order to try to little improve this good revision.
1) Neither figure nor tables are cited along the text.
2) The paragraph between lines 66 and 74 seems to have a different letter format.
3) In section: BCG vaccine: Immunometabolic Reprogramming and Trained Immunity will be more easy to understand and see in a complete vision if author include a figure
4) In the same way in Novel TB vaccines section a table will be recommended.
Author Response
Only minor point is required in order to try to little improve this good revision.
- Neither figure nor tables are cited along the text.
Response: Thanks for noting this. We apologize for the mistake. In the updated version we have cited both figures and tables in the text.
2) The paragraph between lines 66 and 74 seems to have a different letter format.
Response: Thanks for noting this. Sorry for the confusion. It was the caption of the figure and therefore it was a different font style, which might get disturbed during the formatting according to the journal’s style.
3) In section: BCG vaccine: Immunometabolic Reprogramming and Trained Immunity will be more easy to understand and see in a complete vision if author include a figure
Response: Thanks for the suggestion. We agree and therefore included another figure in the manuscript. Kindly check figure 2. We have included detailed metabolomic and epigenetic changes during BCG-induced trained immunity.
4) In the same way in Novel TB vaccines section a table will be recommended.
Response: Thanks for the suggestion. We have included a table containing novel TB vaccine candidates indicating their clinical development status (check table 2).
Round 2
Reviewer 2 Report
Considering corrections made by the authors, the paper might be recommended for publishing.
Reviewer 3 Report
Accept in present form